# Selected Functions and Disorders of Mitochondrial Metabolism under Lead Exposure

**DOI:** 10.3390/cells13141182

**Published:** 2024-07-11

**Authors:** Mikołaj Chlubek, Irena Baranowska-Bosiacka

**Affiliations:** Department of Biochemistry and Medical Chemistry, Pomeranian Medical University, Powstańców Wlkp. 72, 70-111 Szczecin, Poland; mikolaj.chlubek@gmail.com

**Keywords:** lead, mitochondria, energy metabolism, apoptosis, necrosis, autophagy, mitophagy, mitochondrial dynamics, oxidative stress, inflammation

## Abstract

Mitochondria play a fundamental role in the energy metabolism of eukaryotic cells. Numerous studies indicate lead (Pb) as a widely occurring environmental factor capable of disrupting oxidative metabolism by modulating the mitochondrial processes. The multitude of known molecular targets of Pb and its strong affinity for biochemical pathways involving divalent metals suggest that it may pose a health threat at any given dose. Changes in the bioenergetics of cells exposed to Pb have been repeatedly demonstrated in research, primarily showing a reduced ability to synthesize ATP. In addition, lead interferes with mitochondrial-mediated processes essential for maintaining homeostasis, such as apoptosis, mitophagy, mitochondrial dynamics, and the inflammatory response. This article describes selected aspects of mitochondrial metabolism in relation to potential mechanisms of energy metabolism disorders induced by Pb.

## 1. Introduction

Mitochondria play a fundamental role in the energy metabolism of eukaryotic cells. The origin of these organelles is explained by the endosymbiosis theory, according to which proteobacteria were engulfed by cells and fully integrated into the host metabolism [1]. The primary function of mitochondria is the synthesis of high-energy ATP molecules through oxidative phosphorylation (OXPHOS) in relation to glycolysis, oxidative decarboxylation of pyruvate, the tricarboxylic acid cycle (TCA), and β-oxidation of fatty acids [2]. Additionally, mitochondria serve as crucial intracellular signaling organelles by producing reactive oxygen species (ROS), participating in inflammation, and regulating cell death [3].

Mitochondria possess their own circular, double-stranded DNA with an organized structure, referred to as the nucleoid [4]. As double-membraned organelles, they form compartments with distinct chemical properties and functions: the intermembrane space (IMS) and the matrix. The outer membrane (OMM) exhibits higher permeability compared to the inner membrane (IMM), which is necessary to maintain the concentration gradient between the matrix and IMS, driving ATP synthase activity. The electron transport chain (ETC) consists of five protein complexes and constitutes the most significant component of IMM, directly participating in gradient generation and oxygen-related processes associated with OXPHOS [5].

Disorders of energy metabolism involving mitochondria are closely related to the pathogenesis of certain diseases, including those of the cardiovascular, nervous, immune, renal, and pulmonary systems [6]. Numerous studies indicate lead (Pb) as a widely occurring environmental factor (contaminated soil, tap water in homes that have lead pipes, accumulators, aviation fuel, lead-based paint) capable of disrupting oxidative metabolism by modulating the mitochondrial apparatus. Easily absorbed through the respiratory tract, gastrointestinal system, and even the skin, Pb primarily accumulates in bones, although it can readily reach all vital organs, including the brain. Using mainly voltage-dependent calcium channels (VDCCs), lead easily penetrates into cells. It can completely alter the intracellular environment by blocking certain calcium-binding proteins (e.g., calmodulin, PKC). Another well-known mechanism of lead toxicity is the displacement of zinc as a protein cofactor. This occurs, for example, in erythrocytes, where lead can inhibit delta-aminolevulinic acid dehydratase (ALAD), a key enzyme involved in heme synthesis. Impaired N-methyl-D-aspartate (NMDA) receptor-mediated glutamate neurotransmission and dysfunctional zinc finger domains of some transcription factors also result from lead mimicking zinc. Interference with iron metabolism by competing for access to iron transport proteins (e.g., DMT-1) is a less well-studied, but nonetheless important, aspect of lead toxicity. In conclusion, the multitude of known molecular targets of Pb and its strong affinity for biochemical pathways involving divalent metals (Ca, Fe, Cu, Zn, Mn) suggest that it may pose a health threat at any given dose [7,8]. This article describes selected aspects of mitochondrial metabolism in relation to potential mechanisms of energy metabolism disorders induced by Pb.

## 2. Energy Supply—A Crucial Function of Mitochondria

Maintaining homeostasis in eukaryotic cells depends greatly on the availability of ATP, which is synthesized during glycolysis and oxidative phosphorylation. Comparing the energy balance of both processes, it is easy to see the overriding importance of mitochondrial oxidative metabolism in meeting the advanced needs of tissue organisms. The continuity of mitochondrial ATP synthesis is mainly maintained through the oxidation of metabolites in the tricarboxylic acid (TCA) cycle and β-oxidation of fatty acids, occurring in the matrix. During these processes, the regeneration of NADH and FADH_2_, reduced forms of nucleotides, occurs, supplying electrons, respectively, to complex I (NADH-coenzyme Q reductase) and complex II (succinate-ubiquinone reductase) of the electron transport chain (ETC). Electron transfer between successive components of the ETC links oxygen reduction with proton movement across the inner mitochondrial membrane, resulting in the existence of an electrochemical proton gradient between the matrix and the intermembrane space [9]. An intermediate effect of these transformations is the synthesis of reactive oxygen species (ROS) [10]. The free flow of protons back through the channel of complex V (ATP synthase) allows for the generation of ATP from ADP and P_i_ [11].

### Inhibition of ATP Generation in the Cell by Lead

Changes in the bioenergetics of cells exposed to Pb have been repeatedly demonstrated in research, primarily showing a reduced ability to synthesize ATP. This is of paramount importance for the fate of the nervous system—an environment particularly sensitive to energy deficits. The mechanisms of energy disturbances include the inhibition of the electron transport chain (ETC), alterations in the activity of enzymes involved in glucose metabolism, inhibition of Na^+^/K^+^-ATPase, and a decrease in mitochondrial membrane potential [12,13,14,15].

In cerebellar neurons obtained from offspring of rats prenatally exposed to subclinical doses of Pb (PbAc oral administration during pregnancy resulted in a concentration of 5–10 μg/dL in the offspring blood), changes in the cellular energy status have been observed, including a decrease in ATP and total adenine nucleotide (TAN) concentrations, as well as a reduction in the adenylate energy charge (AEC) value. Lead has been observed to have an inhibitory effect on the activity of Na^+^/K^+^-ATPase as well as increased ROS production in cells [12].

In the brain tissue of rats treated with lead acetate, enzymes involved in glucose metabolism were investigated. Reduced activity of pyruvate dehydrogenase complex (PDHC) and hexokinase (HK) was demonstrated. It is worth noting that supplying substrates (glucose, ATP) to cells preserved hexokinase activity, suggesting resistance to the action of Pb after substrate binding [13].

Changes in glucose metabolism due to short-term exposure to lead have been observed in mouse kidney cells. A significant decrease in the activity of glucose-6-phosphatase and 3-phosphoglycerate aldehyde dehydrogenase was described, suggesting disruptions in glycolysis. In addition to this observation, the study reported stimulation of TCA enzymes—malate dehydrogenase and succinate dehydrogenase [14].

In isolated human monocytes, the toxic effect of lead on energy metabolism results in impaired phagocytic ability [15]. It inhibits the activity of ETC complexes I and III and reduces ATP production in cells. It is possible that coenzyme Q10 exerts a protective effect, preserving phagocytic activity and counteracting energy disturbances [15].

## 3. Apoptosis and Necrosis

The development of multicellular organisms is governed by cyclically progressing processes of cell division, differentiation, maturation, and death. The elimination of cells from the pool of biologically active entities can occur under various circumstances, with apoptosis and necrosis being the best-known pathways [16]. 

Necrosis most commonly occurs due to the overwhelming impact of external factors, resulting in mechanical damage to the cell. Necrosis is a pathological process that occurs passively, without the involvement of specific signaling pathways [17]. Its causes may include extreme temperatures, UV radiation, and environmental toxins, including Pb [18]. As a result of overload and ultimately the breakdown of compensatory mechanisms, the lipid bilayers of cell membranes become highly permeable, leading to an uncontrolled influx of ions and water from the external environment. Progressive organelle swelling and metabolic inhibition ultimately result in the loss of membrane integrity and the release of cell contents. Specific metabolites present in the extracellular space usually initiate an inflammatory response aimed at limiting the extent of damage and restoring homeostasis [19].

In contrast to necrosis, apoptosis is a physiological, purposeful, and tightly controlled process. It requires active involvement of cellular machinery, expression of multiple genes, and expenditure of energy. The existence of a programmed cell death mechanism in a healthy organism is biologically justified because it allows for the precise termination of defective cells, potentially capable of further division and replication errors [20]. The first visible change indicating the onset of apoptosis is the condensation of nuclear chromatin, which then migrates to the vicinity of the nuclear membrane. In subsequent stages, nuclear fragmentation, cytoplasmic condensation, and vesicle formation occur. Ultimately, the cell disintegrates into apoptotic bodies—small membrane-bound structures containing chromatin fragments and remaining organelles—which are then engulfed by phagocytic cells [21]. Due to the complete isolation of the intracellular environment from the extracellular milieu during apoptosis, there is no inflammatory process [22].

At the molecular level, there are several pathways of apoptosis initiation, among which the intrinsic pathway is directly dependent on mitochondria. In this variant, the initiating stimulus comes from within the cell and may include DNA damage, for example [23]. Detected abnormalities trigger the expression of the transcription factor p53, which then engages a series of proapoptotic genes for the synthesis of regulatory proteins, with particular emphasis on the B-cell CLL/lymphoma 2 (Bcl-2) family [24]. While Bcl-2 and Bcl-xL proteins function as anti-apoptotic factors, Bax protein and its relative Bak serve as key proapoptotic components. Bax is activated by BH3 domain-containing proteins, such as Bim, tBid, and p53 upregulated modulator of apoptosis (PUMA) [25,26]. Presumably, another BH3 domain-containing protein, Bad, binds to Bcl-2/Bcl-xL, limiting their function [27]. 

Optimally functioning Protein Kinase B (AKT) in the cell prevents the initiation of proapoptotic processes at multiple levels, including direct interaction with Bax, inhibition of Bad, and inhibition of GSK3b kinase, which promotes mitochondrial localization of Bax. In a dying cell, AKT availability significantly decreases, and processes are directed toward apoptosis [28,29]. Active Bax is translocated to the outer mitochondrial membrane, where it undergoes oligomerization and simultaneously forms relatively large pores, allowing the transport of intermembrane space contents to the cytoplasm [30].

The released factors include cytochrome c, Smac/Diablo, and HtrA2/omi proteins [31]. The latter two bind to antiapoptotic inhibitors of apoptosis proteins (IAPs), inhibiting their activity [32,33]. The most significant factor originating from the intermembrane space, however, is cytochrome c, which, along with apoptotic peptidase activating factor 1 (Apaf-1) and pro-caspase 9, forms a protein complex in the cytoplasm called the apoptosome. The role of the apoptosome is to activate executioner caspases, enzymes with broad proteolytic properties, whose presence is a specific indicator of apoptosis progress [34] (Figure 1).

Until recently, mitochondrial permeability transition pores (MPTPs) located in the regions connecting the inner and outer mitochondrial membranes have been attributed particular significance in the intrinsic apoptosis pathway. MPTPs are large protein complexes composed of, among others, adenine nucleotide translocase (ANT), voltage-dependent anion channel (VDAC), and peripheral benzodiazepine receptor (PBR) [35,36]. Both cytosolic proteins (Bcl-2, Bax) and matrix proteins (Cyp-D) can interact with MPTPs [37]. In the resting state, MPTPs exhibit low selectivity but limit the flow of molecules larger than 1.5 kDa (kilodaltons). In response to stress stimuli, such as excessive ROS production, oxidative damage, or increased Ca^2+^ concentration in the cytoplasm, they open, which is undoubtedly associated with cell commitment to termination. Bax-dependent opening of MPTPs leads to irreversible changes in membrane permeability (permeabilization), resulting in metabolic disturbances, mitochondrial swelling, and release of apoptogenic factors (cytochrome c, Smac/Diablo, HtrA2/omi + apoptosis inducing factor, and endonuclease G) [38,39]. Both endonuclease G and AIF are transported to the nucleus, where they participate in DNA degradation [40,41] (Figure 2).

Considering the involvement of open MPTPs in promoting inflammation (e.g., NLRP3 inflammasome activation and DAMPs—damage-associated molecular pattern release), it is difficult to conclusively determine whether this process solely supports late apoptosis or rather converts it to necrosis. For example, in the case of reperfusion syndrome, where ATP availability is insufficient for proper apoptosis processing, permeabilization may contribute to the generation of necrotic changes [42].

### Lead as a Modifying Factor in the Cell Death

Lead, as a toxin interacting with numerous proteins, does not remain inert towards factors regulating the intrinsic apoptotic pathway. Exposure to Pb evidently leads to an imbalance between antagonistically acting proteins of the Bcl-2 family. In the presence of Pb, the level of Bax protein significantly increases, while the expression of Bcl-2 and Bcl-xL proteins decreases [43,44]. Early changes most commonly involve increased synthesis of p53 protein, while the proapoptotic effect is defined by activated caspases. 

Administering low doses of Pb to rats for 40 days resulted in a significant increase in the Bax/Bcl-2 ratio and increased the activity of executioner caspase 3 in cardiovascular tissues [43]. In rabbits fed lead acetate, high oxidative stress markers and systemic tissue changes were observed, correlating with increased expression of p53 protein and decreased expression of Bcl-2 protein [45]. Similar results were obtained by studying the effect of Pb on mouse skin fibroblasts, where increased expression of p53, Bax, caspases 3, 8, and 9, and decreased Bcl-2 activity were observed. Moreover, there was an increase in intracellular calcium ion concentration, intensification of ROS production, and a decline in mitochondrial membrane potential [44]. These and many other studies provide evidence of the proapoptotic action of Pb, both through direct interference with the apoptotic regulatory system and broader disruptions of intracellular homeostasis. A strict association between the intensity and extent of apoptotic changes with the duration of exposure and the administered dose of Pb has also been proposed.

Considering the involvement of MPTP channels in cell death, they should be considered as one of the targets sensitive to Pb intoxication. Ma et al. investigated a series of mitochondrial function disorders in rat liver cells exposed to Pb. They observed a decrease in ATP synthesis, associated with inhibition of electron flow chain complexes, particularly complex III. In addition to the overproduction of mitochondrial ROS, they also found mitochondrial membrane permeabilization, destabilization of its structure, loss of membrane potential, MPTP opening, and mitochondrial swelling followed by cell death [46]. 

An association between Pb intoxication and forced MPTP opening has also been presented in relation to neuronal cells. Stimulating conformational changes in MPTPs itself was closely associated with intensified synthesis of cyclophilin D (Cyp-D), a matrix regulatory protein. The study proposed cyclosporine A (CSA) as a mitigating factor for the destructive effects of Pb presence in mitochondria. Inhibition of Cyp-D caused by CSA administration allowed for limiting MPTP opening, reducing oxidative stress markers, and protecting mitochondria from swelling and rupture [47]. 

Liu et al. observed identical consequences of Pb presence in proximal tubule cells of rat kidneys. The study demonstrated the involvement of MPTP opening in progressive mitochondrial dysfunction, where impairment of ATP synthesis by Pb was the initiating factor for changes. The authors also examined the effects of selected inhibitors of MPTP component molecules, suggesting their beneficial effects in inhibiting Pb-induced cell death [48].

## 4. Mitophagy, Mitochondrial Dynamics, and Quality Control

During the course of evolution, eukaryotic cells developed a variety of mechanisms to control the optimal progression of metabolic processes. One of the more radical processes sustaining the quality of cellular components is the degradation of its own resources in a process called autophagy [49]. This evolutionarily conservative phenomenon, discovered in the mid-20th century, has been classified as a type of cell death and, depending on the scale of degradation, occurs in three variants: macroautophagy, microautophagy, and chaperone-mediated autophagy (CMA). 

Macroautophagy involves the isolation of entire organelles along with a portion of the cytoplasm in de novo-formed membranous structures called autophagosomes. In mammals, recruited autophagosomes are directed to lysosomes, where the fusion of membranes leads to the release and subsequent digestion of contents by hydrolase enzymes [50]. Microautophagy refers to cytosolic molecules that are directly targeted to lysosomes and undergo degradation [51]. The last type of autophagy, CMA, concerns specific peptides marked by the KFERQ sequence, selectively recognized and translocated to lysosomes with the assistance of chaperone proteins [52]. The full significance of autophagy remains unclear; however, it can be hypothesized that the process aims to eliminate dysfunctional and inefficient organelles, and at the molecular level, enables the utilization of misfolded or damaged proteins. An important aspect of autophagy is resource recovery and the possibility of their reuse by the cell, which seems particularly significant under conditions of limited access to nutrients [53].

Mitochondria form a well-connected network structure within the cell, which undergoes remodeling depending on the cell’s energy demand. Changes in the quantity and morphology of mitochondria are possible due to alternately occurring fusion and fission processes, collectively referred to as mitochondrial dynamics [54]. Fusion allows for the exchange of resources between mitochondria, significantly enhancing ATP synthesis efficiency and minimizing the consequences of component damage, such as mitochondrial DNA. The process occurs in two stages and requires the involvement of specific proteins from the GTPase family. Initially, the outer mitochondrial membranes are linked through mitofusin 1 (MFN1) and mitofusin 2 (MFN2). Then, the OPA1 protein mediates the fusion of inner membranes. Fusion predominates in highly metabolically active cells, such as neurons [55,56]. Fragmentation, on the other hand, dominates in cells with low energy demand and involves breaking down the mitochondrial network into many small organelles. The main proteins involved in this process are dynamin-related protein 1 (DRP1) and mitochondrial fission factor (MFF). Besides its role in cellular bioenergetic adaptation, mitochondrial fragmentation presumably allows for the isolation of damaged structures from the active mitochondrial apparatus [57,58].

One of the factors mediating mitochondrial network shaping is the peroxisome proliferator-activated receptor gamma coactivator 1α (PGC-1α). As a protein that is an important regulator of mitochondrial biogenesis and dynamics, PGC-1α interacts with a number of transcription factors (NRF1, NRF2, and ERRα), promoting an increase in the number of mitochondria and the synthesis of proteins involved in oxidative phosphorylation [59,60,61]. PGC-1α has a protective effect on cells sensitive to energy deficits, for example, by increasing the participation of antioxidant enzymes in ROS detoxification. Optimal PGC-1α expression probably significantly limits the progression of destructive changes in neurodegenerative diseases [62].

The fundamental tool involved in mitochondrial quality control is the ubiquitin–proteasome system (UPS) [63]. Specific ligases present in the outer mitochondrial membrane mark specific mitochondrial proteins by binding them to ubiquitin monomers or polymers. Prepared proteins are directed to cytoplasmic proteasomes, where they undergo degradation. Minor deviations from the physiological state may be fully reversible due to protease activity, antioxidant enzymes, and DNA repair proteins. A high degree of disruption of mitochondrial homeostasis necessitates the selective autophagy of dysfunctional organelles, referred to as mitophagy [64]. 

The main pathway promoting mitophagy in mammalian cells is based on the relationship between PINK1 and PARKIN proteins [65,66]. Under physiological conditions, PINK1 kinase enters mitochondria, where it interacts with translocases of both mitochondrial membranes (TOM and TIM). PINK1 localization in the membrane area initiates the N-terminal modification by inner membrane enzymes [67]. 

Initially, a specific peptidase MMP cleaves the MTS fragment, determining PINK1 affinity to mitochondria, and then the PARL protease removes the F104 sequence. Processed PINK1 undergoes ubiquitination and shortly afterward proteasomal degradation; its short lifespan reflects the optimal state of mitochondria [68] (Figure 3). 

The situation changes when PINK1 reacts with the depolarized membrane of damaged mitochondria. By forming a complex only with TOM, PINK1 is embedded, stabilized, and self-activated in the outer mitochondrial membrane. In this variant, PINK1 initiates the mitophagy cascade by phosphorylating ubiquitin and the UBL domain of PARKIN, an E3 ubiquitin ligase of cytosolic origin [69]. Processed PARKIN, in cooperation with phosphorylated ubiquitin, becomes activated and carries out numerous ubiquitinations of mitochondrial proteins. Depending on the amino acid residue location to which PARKIN transfers ubiquitin, proteins may undergo proteasomal degradation or participate in communication with adaptor proteins [70,71] (Figure 3). Upon detection and installation in regions rich in polyubiquitin chains, adaptors facilitate the binding of receptors from the light chain protein 3 (LC3) family, characteristic of the inner surface of the autophagosome membrane. Proteins such as OPTN, NDP52, or p62 thus condition the possibility of coating and preparing marked mitochondria for degradation [72,73].

There are also alternative initiation pathways of mitophagy, independent of the PINK1/PARKIN system. They involve direct binding of LC3 by specific mitochondrial membrane receptors. The types of discovered receptors seem to be tissue-specific and constitute a complement and/or alternative pathway for the inefficient PINK1/PARKIN pathway [74,75].

### Lead-Induced Disruptions in Mitochondrial Dynamics and Mitophagy 

As we better understand the complexity and importance of processes responsible for maintaining mitochondrial homeostasis, recent years have seen research into the impact of Pb on this delicate ecosystem. Pb not only disrupts cellular oxidative metabolism by directly damaging mitochondria but also suppresses mechanisms related to their repair, biogenesis, and dynamics. Numerous publications unequivocally indicate lead’s involvement in the etiology of neurodegenerative diseases, emphasizing the brain’s particular vulnerability to disruptions in energy metabolism.

Han et al. described a series of disturbances in chicken spleen cells subjected to Pb exposure. In addition to increased levels of pro-inflammatory cytokines and oxidative stress markers, the authors demonstrated changes in the expression of genes involved in mitochondrial dynamics. Increased involvement of MFF and DRP1, along with decreased activity of MFN1, MFN2, and OPA1, indicated lead’s promotion of mitochondrial fragmentation. Oxidative damage and difficulties in maintaining mitochondrial network integrity were likely causes of increased autophagy in cells. The study yielded promising therapeutic effects using selenium (Se), which reduced oxidative stress levels, supported mitochondrial dynamics, and prevented autophagy [76]. 

Yang et al. confirmed the relationship between Pb-induced oxidative stress and disturbances in mitochondrial dynamics. Using a neuroblastoma cell line, the authors described ROS-dependent activation of DRP1, initiating fragmentation. The study provided valuable insights into the beneficial effects of metformin on lead-associated oxidative stress control. Metformin’s protective action involved increased expression of the NRF2 factor, which mediates, among other things, the synthesis of antioxidant enzymes [77].

Recently, researchers have increasingly emphasized the significance of the PGC-1α factor as the main regulator of mitochondrial biogenesis and dynamics. Dabrowska et al. pointed to the significant involvement of PGC-1α in protecting neurons from the effects of Pb poisoning. The profile of PGC-1α action is in natural opposition to some of the disruptions caused by Pb, which provides an argument for deepening our knowledge of the importance of this protein in the prevention of neurodegenerative diseases. Interestingly, the authors suggested that both insufficient and excessive expression of PGC-1α may increase the risk of mitochondrial dysregulation by lead [78].

Pb can modify autophagy both in its general scope and at the level of specific pathways related to mitophagy, although many issues in this area remain unclear, and the conclusions proposed in studies are sometimes contradictory.

In a study devoted to the nephroprotective role of heme oxygenase 1 (HO-1), it was shown that Pb blocks autophagy by inhibiting the expression of LC3-II and Beclin 1, key proteins for this process. The authors found that overexpression of HO-1 allowed the resumption of autophagy function, thereby protecting cells from apoptosis [79]. Similar results were achieved by Liu et al., describing autophagy defects in neuronal cells of fish embryos exposed to various doses of Pb. A decrease in the levels of LC3-II and Beclin 1 was demonstrated, resulting in autophagy arrest and shortening of embryo lifespan. Partial reversal of neurotoxic effects was achieved by using rapamycin [80]. Meanwhile, Han et al. observed increased expression of genes related to autophagy in chicken spleen cells. Pb-stimulated autophagy was evidenced by elevated levels of Beclin 1, Dynamin, Atg 5, LC3-I, and LC3-II proteins, as well as decreased expression of the mammalian target of rapamycin (mTOR) [76].

The PINK/PARKIN pathway is an obvious direction for research into lead’s involvement in modulating mitophagy, considering the disturbances in mitochondrial membrane potential induced by Pb-exposed mitochondria. Gu et al. proposed a mechanism in which Pb indirectly accelerates mitophagy by increasing ataxia telangiectasia mutated kinase (ATM) expression, a protein closely associated with PINK1 activity. ATM participates in the phosphorylation of PINK1 and PARKIN, potentially amplifying the ubiquitination process of mitochondrial proteins. The authors demonstrated a decrease in the expression of mitochondrial markers, such as cytochrome c oxidase IV (COX IV) and heat shock protein 60 (HSP60), illustrating the progressive elimination of mitochondria under Pb exposure conditions [81]. In a recent study on neuronal cells, impairment of mitophagy under Pb and β-amyloid protein influence was observed. The presence of stressors resulted in significantly reduced levels of PINK1 and PARKIN. Mitophagy arrest led to the accumulation of dysfunctional mitochondria and forced cell apoptosis [82]. 

Gao et al. suggested a regulatory role of the endoplasmic reticulum (ER) in mitophagy. ER subjected to stress (Pb) promoted mitochondrial degradation, while reducing stress halted this process. The study observed increased expression of LC3-II and decreased expression of the mitochondrial marker HSP60, indicating lead’s stimulating effect on mitophagy [83].

The reported research results do not unequivocally predict changes in mitochondrial metabolism under the influence of Pb. However, they show a clear correlation between exposure and the induction of disturbances in the described processes. Further studies on the impact of Pb on mitochondrial dynamics and mitophagy may help better understand the origins of neurodegenerative diseases, given the key role of energy metabolism in neuron health and functionality.

## 5. Mitochondria and Oxidative Stress

An inherent phenomenon accompanying physiological processes occurring within mitochondria is the production of reactive oxygen species (ROS). The generation of ROS can be explained by the properties of oxygen, which in its molecular form contains two unpaired electrons with parallel spins. Such a configuration of electron shells does not favor the simultaneous acceptance of a pair of electrons, meaning molecular oxygen must be gradually reduced. Acceptance of a single electron results in the formation of the superoxide anion radical (O_2_^•−^), which is primarily generated as a byproduct of the oxidation of NADH and FADH_2_ at complexes I and III of the respiratory chain [84,85]. 

To prevent the toxic effects of accumulation, O_2_^•−^ undergoes spontaneous conversion with the involvement of superoxide dismutase (SOD). Importantly, O_2_^•−^ is generated on both sides of the inner mitochondrial membrane, and its limited ability to diffuse through this barrier explains the presence of various SOD variants in the matrix and intermembrane space. In a reaction catalyzed by SOD, oxygen and hydrogen peroxide (H_2_O_2_) are formed, the latter also classified as a ROS [86]. Although H_2_O_2_ is not a radical, it serves as a potential substrate for the Fenton/Haber–Weiss reaction, in which the highly reactive hydroxyl radical (HO^•^) is generated concurrently with the oxidation of metal ions, mainly Fe^2+^ and Cu^2+^ [87,88]. Further reduction of H_2_O_2_ to H_2_O and O_2_ is possible due to the presence of specific enzymes such as catalase (CAT), nonspecific enzymes like glutathione peroxidase (GPX) and thioredoxin peroxidase (TPX), other non-enzymatic antioxidants such as glutathione (GSH), and vitamins C and E [89].

Under physiological conditions, a state of equilibrium is established between ROS production and the antioxidant system. Radicals maintained at low concentrations participate in cellular signaling processes. In phagocytic cells, ROS synthesis and accumulation act as a chemical weapon against pathogens. The physiological role of ROS has been demonstrated, for example, in granulomatous diseases, where impaired NADPH oxidase function and reduced O_2_^•−^ synthesis resulted in a significantly higher susceptibility to infections among those studied [90]. On the other hand, disturbance of redox balance through ROS overproduction and/or decreased antioxidant efficiency leads to oxidative stress (OS), posing a threat to intracellular macromolecules. 

Lipid and lipoprotein peroxidation, especially of polyunsaturated fatty acids sensitive to oxidation, manifests as damage to cell membranes. In the case of mitochondria, increased membrane permeability and inadequate induction of apoptosis may occur [91]. Lipid peroxidation also leads to the synthesis of free malondialdehyde (MDA), a cytotoxic and potentially mutagenic compound, which serves as the most sensitive biochemical marker of OS [92,93]. Structural aberrations of structural and enzymatic proteins may occur, leading to changes in conformation and disruptions or loss of their function. There are also aberrations in nucleic acids, evidenced by the formation of, among others, 8-hydroxy-2-deoxyguanosine (8-OHdG), representing one of the better-understood indicators of oxidative DNA damage [94].

### Involvement of Lead in Inducing Oxidative Stress

Chronic poisoning even with small doses of Pb induces oxidative stress, both by promoting reactive oxygen species (ROS) production and modulating antioxidant activity [95]. Electron flow in the electron transport chain (ETC) slows down in the presence of xenobiotics, including Pb^2+^. This leads to an increase in the NADH/NAD^+^ ratio and intensification of oxygen reduction to O_2_^•−^ [46]. Antioxidant enzymes are targeted by Pb^2+^ as proteins requiring specific cofactors to perform their functions. Among the known variants of superoxide dismutase (SOD), Cu^2+^/Zn^2+^-dependent SOD-1 is present in the cytosol and mitochondrial intermembrane space, while Mn^2+^-dependent SOD-2 is exclusively present in the matrix. Pb^2+^ can compete with all aforementioned metal ions for binding to prosthetic centers, thereby disrupting the structural stability and the ability of SOD to process O_2_^•−^ [96].

The antioxidant potential of glutathione (GSH) arises from the presence of thiol groups, which are a direct target for Pb ions. The binding of Pb^2+^ to -SH groups results in a noticeable decrease in GSH levels and reduces the participation of this tripeptide in the antioxidant system [97]. Additionally, glutathione reductase (GSSGR), an enzyme responsible for the regeneration of GSH from its oxidized form (GSSG), is also inhibited by Pb^2+^.

The impact of Pb^2+^ on the activity of antioxidant enzymes has been addressed in many studies. Results generally suggest the existence of the limited protective potential of enzymes, depending on the degree of Pb^2+^ poisoning. In plant cells subjected to short-term (8 h) exposure to Pb ions, a concentration-dependent abrupt increase in SOD activity and other antioxidant enzymes was observed as a response to intense O_2_^•−^ and H_2_O_2_ production. The importance of CAT in cell protection at the early stage of intoxication was shown to be minor [98]. In another study on plant cells, a significant increase in CAT activity was observed three days after the application of Pb salts, which was a consequence of earlier H_2_O_2_ overproduction, among others, in mitochondria, leading to a reduction in OS parameters induced by Pb^2+^. Moreover, high activity of various SOD variants in the cytosol, mitochondria, and peroxisomes was demonstrated [99]. In an experimental study on rats, a decrease in SOD and CAT activity was observed in brain tissue cells exposed to Pb^2+^. This phenomenon was explained by disturbances in copper and iron metabolism caused by the presence of Pb^2+^, which affected the synthesis of the aforementioned enzymes [100]. Similar results were obtained in a study on mouse testicular cells exposed to lead, where a decrease in SOD, CAT, and GPX activity was correlated with reduced gene expression of these proteins [101]. The degree of expression of different SOD subtypes depending on the Pb^2+^ concentration in mite cells was also described. It was observed that SOD activity initially increased and then decreased with increasing Pb^2+^ concentration but remained significantly higher than in the control group without Pb exposure [102].

Studies on human populations exposed to Pb confirm the relationship between blood lead levels (BLLs) and the level of selected oxidative stress biomarkers. MDA was often chosen as a key parameter, and multiple studies have shown proportionally high MDA values with higher BLLs, especially in populations at occupational risk [103,104]. Elevated BLLs usually resulted in decreased GSH levels, and moreover, simultaneous increases in its oxidized form (GSSG) were indicated [105,106,107]. Interesting results were presented after examining a group of occupationally exposed workers, among whom individuals with relatively lower BLLs had an increased GSH index, suggesting active compensation for OS. With higher BLLs, GSH reserves were likely depleted [108].

Oxidative stress induced by exposure to Pb carries systemic consequences for human health. The nervous system appears to be particularly susceptible to damage, as evidenced by the inclusion of OS in the etiology of neurodegenerative diseases such as Alzheimer’s disease or Parkinson’s disease [109,110]. Oxidative damage to nucleic acids can lead to the activation of oncogenes and chromosomal aberrations resulting in carcinogenesis [111]. In the mechanism of OS-induced inflammatory processes, numerous pathologies affecting the nervous, cardiovascular, renal, articular, and pulmonary systems have been documented [6,90].

## 6. Mitochondria and Inflammation

Inflammation is a complex and multifactorial process involving the body’s defense mechanisms. Under physiological conditions, it encompasses the coordinated cooperation of immune system cells with cells exposed to pro-inflammatory factors of external origin and/or tissue damage. Depending on the cause, duration, extent, and intensity, inflammation can have extremely different health outcomes, ranging from effective pathogen elimination and tissue regeneration to cell necrosis and chronic disease states [112]. Mitochondria constitute a significant element of the molecular infrastructure participating in the inflammatory response [113,114]. Additionally, numerous studies examine the role of mitochondria in close association with innate immune mechanisms [115,116].

Cascade signaling pathways are triggered by the activation of pattern recognition receptors (PRRs). In conditions of infection, various microbial antigens serve as ligands for PRRs and are collectively referred to as pathogen-associated molecular patterns (PAMPs) [117]. Conversely, endogenous compounds binding to PRRs, released from damaged cells and/or subjected to stressors, are termed damage-associated molecular patterns (DAMPs) [118].

Mitochondria mediate signaling pathways of the inflammatory process at multiple levels. Specific mitochondrial macromolecules, which under healthy conditions remain isolated from the intracellular and extracellular environment, upon release function as DAMPs and enhance the pro-inflammatory signal. Mitochondrial DAMPs include, among others, mtDNA, mtROS, ATP, cytochrome c, cardiolipin, and N-formyl peptides [119]. 

In living cells, mtDNA can migrate from the matrix with the involvement of pores associated with proapoptotic proteins Bax and Bak and through open MPTPs. When in the cytoplasm, it activates cyclic GMP-AMP synthase (cGAS), which recognizes double-stranded DNA and functions as a PRR of the nonspecific immune response [120]. Similar to the ligation of foreign nucleic acid (e.g., bacterial), the signal initiated by mtDNA reaches the stimulator of interferon genes (STING) protein. Activated STING translocates from the ER to the vicinity of the nucleus, where it promotes the expression of inflammatory cytokines, including interferon-β1 (IFNβ1), tumor necrosis factor (TNF), and interleukin 6 (IL-6) [121]. Alongside the cGAS-STING pathway, mtDNA participates in NLR Family Pyrin Domain Containing 3 (NLRP3) inflammasome activation, a large protein complex recruiting caspase 1, which is involved in the maturation of interleukins IL-1β and IL-18 [122]. NLRP3 inflammasome recruitment is a result of toll-like receptor (TLR)-associated pathways activation. TLR ligation initiates signal transduction through the adapter proteins MYD88 and/or TIR-domain-containing adapter-inducing interferon-β (TRIF), ultimately leading to nuclear factor kappa-light-chain-enhancer of activated B cells (NF-κB), mitogen-activated protein kinase (MAPK), interferon-regulatory factor 3 (IRF3), and/or IRF7 activation. NF-κB induces the expression of many proteins associated with the inflammatory response, including inactive NLRP3, proIL-1β, and proIL-18 [123]. Mitochondria remain in close relation to the NLRP3 inflammasome, modulating its proteolytic capabilities. Mitochondrial membrane permeabilization promotes NLRP3 and adapter protein apoptosis-associated speck-like protein containing a CARD domain (ASC) relocation to perimitochondrial ER regions. There, the inflammasome assembly occurs through NLRP3 oligomerization, followed by ASC and pro-caspase 1 recruitment. Its activity is determined by, among other factors, mitochondrial DAMPs, especially mtDNA, mtROS, and cardiolipin [124,125] (Figure 4).

The release of mitochondrial molecules into the extracellular space, most commonly occurring with cell death, stimulates cellular immune response mechanisms. In this environment, mtDNA exhibits chemotactic properties by binding to certain PRRs present on neutrophil membranes (TLR9 and AGER—advanced glycosylation end-product receptor), primarily associated with bacterial DNA binding [126]. Strong activators of neutrophils also include N-formyl peptides from the matrix, which utilize the dedicated FPR-1 membrane receptor [127]. Cardiolipin, a phospholipid in the inner mitochondrial membrane, interacts with antigen-presenting cells (APCs), initiating the expression of a CD1d-like molecule. Cardiolipin can also bind to CD1d on the APC surface, stimulating the activation of a specific subset of T lymphocytes, γδ T cells [128]. Finally, the release of high levels of ATP signals the recruitment of APC precursors by interacting with their P2 × 7 membrane receptor [129] (Figure 4).

The diversity of signaling pathways initiated by mitochondrial DAMPs illustrates a low entry threshold for innate immune response upon mitochondrial damage. The manner in which the immune system interprets the export of mitochondrial molecules explains the bacterial origin of organelles absorbed by eukaryotic cells in the endosymbiosis process millions of years ago [130].

### Lead-Induced Mitochondrial Dysfunction as a Cause of an Inflammatory Response

In addition to the tissue pathology associated with Pb exposure, we have to mention adaptive changes in the immune system and the development of inflammatory responses. The functionality of immunologically active cells is modified under the influence of Pb, as evidenced by the intense expression of pro-inflammatory cytokines by macrophages and the impairment of differentiation and development of cells of acquired immunity [115,131].

Considering the role of mitochondria in the model of Pb-dependent inflammatory response, special attention should be paid to TLR signaling. Most mammalian TLRs bind PAMPs and DAMPs, some of which may be activated by mitochondrial-derived DAMPs. One of the effects of TLR activation is the initiation of mitochondria-independent p38 and MAPK pathways, leading to NF-κB expression and the activation of genes associated with the inflammatory response [113,132]. 

In addition to this process, TLRs can initiate a signal involving the TNF receptor-associated factor 6 (TRAF6), which translocates to mitochondria. There, TRAF6 ubiquitinates the electron transport chain (ETC) complex I-associated protein called evolutionarily conserved signaling intermediate in Toll pathways (ECSIT), resulting in the recruitment of mitochondria to phagosomes and a significant increase in ROS production [133]. This signaling variant allows for respiratory bursts in macrophages and conditions their ability to eliminate pathogens [134].

Liu et al. demonstrated inflammation associated with the activation of microglial and astroglial cells in the hippocampus of mice exposed to Pb. The results included elevated levels of IL-1β and TNFα and markers of p38/MAPK and ERK1/2 signaling pathways. The TLR4-MYD88-NF-κB pathway was identified in the study as the most likely route for initiating the inflammatory process [135]. 

Analyzing the effect of low doses of Pb (BLL < 10 µg/dL) on selected brain areas in rats, Chibowska et al. described levels of inflammatory markers in nervous tissue, with a significant increase in cytokines and prostanoids, including IL-1β, IL-6, TGF-β, PGE2, and thromboxane B2 (TXB2). In addition to increased NF-κB transcription factor expression, an increase in the activity of pro-inflammatory enzymes COX-1 and COX-2 was also described [136]. 

Similar results were achieved by Metryka et al. in an in vitro study of THP-1 macrophages exposed to low concentrations of Pb. Cell culture showed macrophage activation under the influence of lead, elevated IL-1β and IL-6 levels, and overexpression of COX-1 and COX-2 [137]. 

In a culture of human A549 cancer cells exposed to lead, a significant increase in NF-κB and aryl hydrocarbon receptor (AhR) transcription factor levels and their target gene products was described, confirming the involvement of corresponding signaling pathways in lead-induced inflammation. The authors also demonstrated an increase in ROS levels and promotion of apoptosis in the examined cells, emphasizing the close relationship between these disorders and inflammation [138]. 

Attiafi et al. induced severe lung inflammation with a clear percentage of apoptotic cells in rats in vivo. The results presented an increase in apoptosis markers, oxidative stress, and inflammatory cytokines, including IL-4, IL-10 and TNF-α. The authors highlighted the significant role of NF-κB and AhR signaling pathways in the genesis of the described changes [139].

## 7. Conclusions

Lead is a commonly present environmental toxin that easily penetrates tissues, even the best-protected ones. The fatally dangerous effects of acute Pb poisoning have long been known, but only in recent decades has awareness increased regarding the seemingly invisible consequences of chronic, subclinical exposure to metal compounds. This article attempts to demonstrate the delicate nature of mitochondria, organelles of fundamental importance both in health and disease, vulnerable to changes induced by Pb. Progress in knowledge about diseases associated with mitochondrial dysfunction justifies the need for the further exploration of Pb molecular targets, the elucidation of mechanisms of disorder induction, and the development of new therapeutic strategies.

## Figures and Tables

**Figure 1 cells-13-01182-f001:**
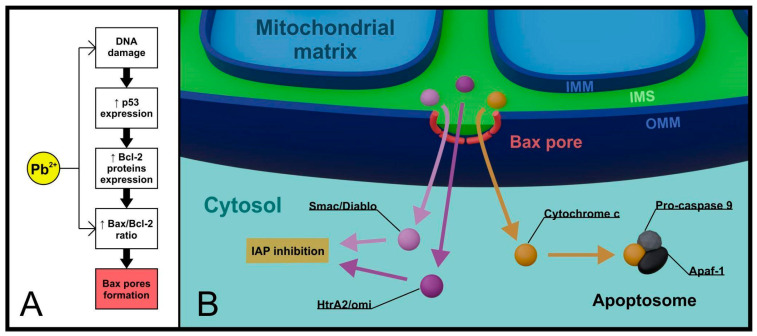
**Lead and mitochondria in the intrinsic pathway of apoptosis.** Critical changes in the cell initiated by lead (e.g., DNA damage) contribute to the activation of the intrinsic pathway. In addition, lead modifies the expression of Bcl-2 family proteins, increasing the importance of the pro-apoptotic protein Bax (**A**). Once it reaches mitochondrial localization, Bax oligomerizes and forms pores in the OMM that allow controlled ejection of apoptogenic factors from the IMS into the cytoplasm (**B**). (IAP—inhibitors of apoptosis proteins, Apaf-1—apoptotic peptidase activating factor 1, IMM—inner mitochondrial membrane, OMM—outer mitochondrial membrane, IMS—intermembrane space).

**Figure 2 cells-13-01182-f002:**
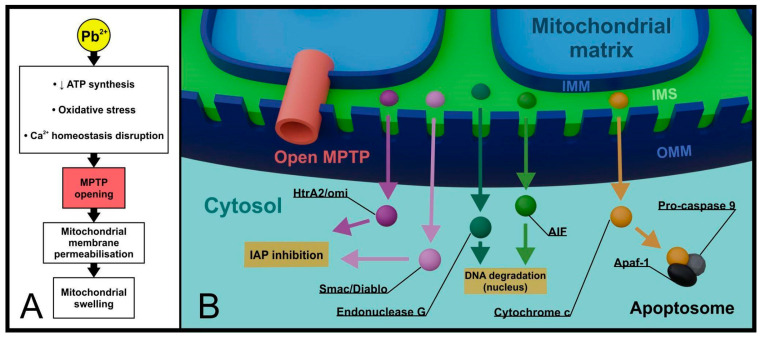
**Lead and MPTP in late apoptosis and necrosis.** Extensive changes in the metabolism of a lead-exposed cell led to the opening of mitochondrial MPTPs. Dramatically increased permeability of the OMM is accompanied by mitochondrial swelling (**A**) and non-selective release of apoptogenic factors from the IMS into the cytoplasm (**B**). (MPTP—mitochondrial permeability transition pore, IAP—inhibitors of apoptosis proteins, AIF—apoptosis inducing factor, Apaf-1—apoptotic peptidase activating factor 1, IMM—inner mitochondrial membrane, OMM—outer mitochondrial membrane, IMS—intermembrane space).

**Figure 3 cells-13-01182-f003:**
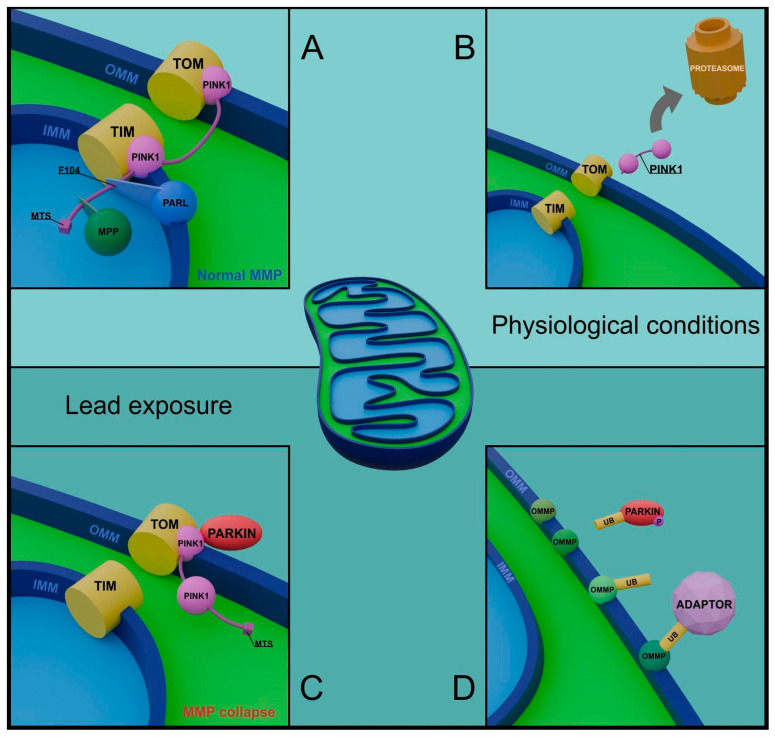
**Role of PINK1 and PARKIN in healthy and damaged mitochondria.** Normal mitochondrial membrane potential (MMP) allows PINK1 to be coupled to the translocases of both membranes (TOM, TIM) and to proceed with changes in the structure of the PINK1 chain (**A**). The processed protein is released into the cytoplasm and degraded in the proteasome, reflecting optimal mitochondrial conditions (**B**). Loss of membrane potential, accompanying pathological changes initiated by lead, determines the permanent binding of PINK1 to OMM and the recruitment of PARKIN protein from the cytoplasm (**C**). Phosphorylated PARKIN performs numerous ubiquitinations of OMM proteins, preparing the damaged mitochondrion for mitophagy (**D**). (TOM—translocase of the outer membrane, TIM—translocase of the inner membrane, PINK1—PTEN induced kinase 1, PARL—PINK1/PGAM5 associated rhomboid-like protease, MTS—mitochondrial targeting sequence, MPP—mitochondrial processing peptidase, MMP—mitochondrial membrane potential, UB—ubiquitin, OMMP—outer mitochondrial membrane protein, IMM—inner mitochondrial membrane, OMM—outer mitochondrial membrane).

**Figure 4 cells-13-01182-f004:**
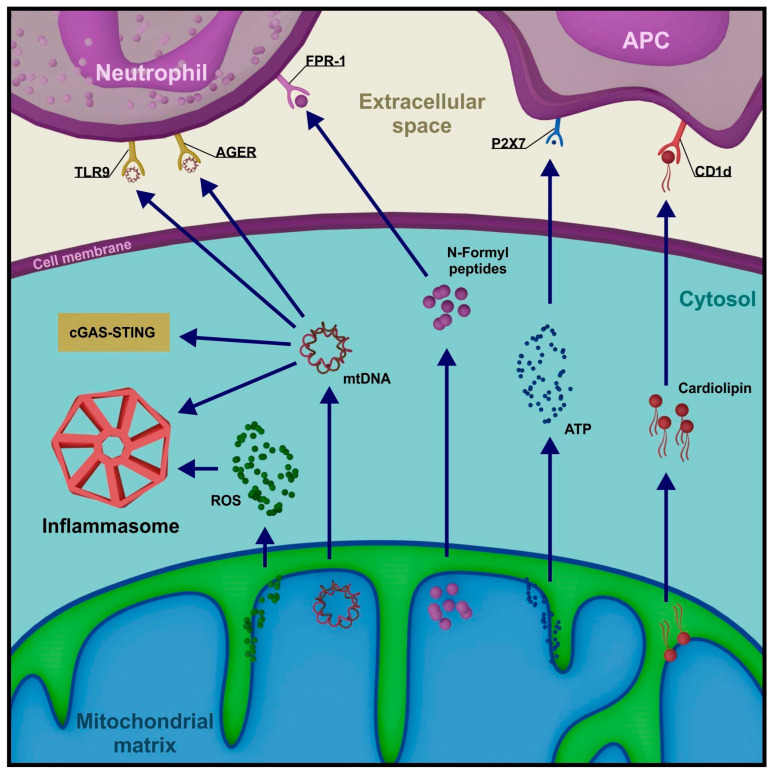
**Selected mitochondrial DAMPs and their role in the amplification of pro-inflammatory signals.** (TLR9—toll-like receptor 9, AGER—advanced glycosylation end-product receptor, FPR-1—formyl peptide receptor 1, CD1d—cluster of differentiation 1d, cGAS—cyclic GMP-AMP synthase, STING—stimulator of interferon genes, ROS—reactive oxygen species).

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
