# Peer review of "Selected Functions and Disorders of Mitochondrial Metabolism under Lead Exposure"

_cells, 2024, doi:10.3390/cells13141182_

Round 1

Reviewer 1 Report

Comments and Suggestions for Authors

Lead (Pb) is a xenobiotic easily absorbed through the respiratory tract, gastrointestinal system, and skin that may pose a health threat at any given dose.

Numerous studies indicate lead (Pb) is capable of disrupting oxidative metabolism and reduces ability to synthesize ATP. Lead also interferes with mitochondrial-mediated processes essential for maintaining homeostasis, such as apoptosis, mitophagy, mitochondrial dynamics and the inflammatory response. 

Chlubek and Baranowska-Bosiacka summarize a few aspects of mitochondrial metabolism in relation to potential mechanisms of energy metabolism disorders induced by Pb.

General comments: 

The topic of the article is extremely interesting and relevant, as knowledge about Pb molecular targets and diseases associated with mitochondrial dysfunction would be crucial to the development of new therapeutic strategies.

The article is clear and well structured, citing recent and relevant publications.

Authors introduce and summarize in a sufficiently detailed and complete way the essential aspects useful to understand the aspects covered regarding the fundamental role mitochondria have in the energy metabolism of eukaryotic cells in relation to potential mechanisms of energy metabolism disorders induced by Pb. 

They also focus on disorders of mitochondrial metabolism essential for the function of these organelles - Apoptosis and Necrosis Mitophagy, Mitochondrial Dynamics, and Quality Control, Oxidative Stress and inflammation - and which appear to be compromised by Pb.

The article also highlights aspects that still need to be clarified or better understood.

Specific comments: 

Only a few minor comments. 

1. Figures summarize and illustrate very clearly the data set out in the text. I would only suggest, if possible, decreasing the size slightly.

2. Lines: 25-27— Authors indicate as primary function of mitochondria the synthesis of high-energy ATP in relation to TCA, β-oxidation of fatty acids. I wonder why they don't include glycolysis as well: glycolysis also produces NADH that will be reoxidized during the electron transport chain in the mitochondrion.

3. Lines: 51-52— The authors refers to ATP synthesized “during glycolysis and oxidative phosphorylation”, but they do not indicate β-oxidation. Is there any reason? I suggest to include β-oxidation.

4. Line 76— The authors should specify the time reference in the administration of the indicated lead dose.

Author Response

Reviewer 1

Lead (Pb) is a xenobiotic easily absorbed through the respiratory tract, gastrointestinal system, and skin that may pose a health threat at any given dose.

Numerous studies indicate lead (Pb) is capable of disrupting oxidative metabolism and reduces ability to synthesize ATP. Lead also interferes with mitochondrial-mediated processes essential for maintaining homeostasis, such as apoptosis, mitophagy, mitochondrial dynamics and the inflammatory response. 

Chlubek and Baranowska-Bosiacka summarize a few aspects of mitochondrial metabolism in relation to potential mechanisms of energy metabolism disorders induced by Pb.

General comments: 

The topic of the article is extremely interesting and relevant, as knowledge about Pb molecular targets and diseases associated with mitochondrial dysfunction would be crucial to the development of new therapeutic strategies.

The article is clear and well structured, citing recent and relevant publications.

Authors introduce and summarize in a sufficiently detailed and complete way the essential aspects useful to understand the aspects covered regarding the fundamental role mitochondria have in the energy metabolism of eukaryotic cells in relation to potential mechanisms of energy metabolism disorders induced by Pb. 

They also focus on disorders of mitochondrial metabolism essential for the function of these organelles - Apoptosis and Necrosis Mitophagy, Mitochondrial Dynamics, and Quality Control, Oxidative Stress and inflammation - and which appear to be compromised by Pb.

The article also highlights aspects that still need to be clarified or better understood.

Specific comments: 

Only a few minor comments. 

  1. Figures summarize and illustrate very clearly the data set out in the text. I would only suggest, if possible, decreasing the size slightly.

Figures have been decreased in size.

  1. Lines: 25-27— Authors indicate as primary function of mitochondria the synthesis of high-energy ATP in relation to TCA, β-oxidation of fatty acids. I wonder why they don't include glycolysis as well: glycolysis also produces NADH that will be reoxidized during the electron transport chain in the mitochondrion.

Thank you for this valuable comment. We have added glycolysis and, additionally, pyruvate decarboxylation that are also generators of NADH, which is further reoxidized in the mitochondrial electron transport chain.

  1. Lines: 51-52— The authors refers to ATP synthesized “during glycolysis and oxidative phosphorylation”, but they do not indicate β-oxidation. Is there any reason? I suggest to include β-oxidation.

In our opinion, the proces of β-oxidation of fatty acids should not be included in this particular statement. Unlike glycolysis, oxidation of fatty acids does not generate ATP (at substrate-level phosphorylation). Instead of this, it provides NADH and FADH2 for oxidative phosphorylation in the electron transport chain. In this statements we took into account two possible circumstances: production of ATP under anaerobic conditions (anaerobic glycolysis exclusively) and its generation under aerobic conditions (oxidative phosphorylation of reducing equivalents produced in glycolysis, pyruvate decarboxylation, the citric acid cycle, and β-oxidation of fatty acids).

  1. Line 76— The authors should specify the time reference in the administration of the indicated lead dose

We added an adequate information in the text (lines 76,77,78)

Reviewer 2 Report

Comments and Suggestions for Authors

The present review provides a detailed description of the molecular mechanisms that affect the dynamics, cell death processes, and alterations that lead to oxidative stress during lead exposure. It also mentions the processes involved and altered by lead in developing diseases, especially in tissues that strictly depend on mitochondrial function. The review is considered adequate for publication in this journal; however, some minor revisions could improve the quality and relevance of this work.

1) On line 42, you could add a paragraph describing the main and most common causes of lead poisoning.

2) In line 51 of point 2, please modify the sentence about metabolism efficiency since, at the end, it talks about glycolysis and oxidative phosphorylation, which are also metabolisms.

3) On lines 71-73, quotes need to support the information mentioned.

4) Along these same lines, it is necessary to justify why emphasis is placed on the impact of lead on the nervous system and its components. 

5) On line 78, what is the significance of AEC’s reduction during cerebellar neurons exposed to lead?

6) On line 79, is the inhibitory effect of lead on Na+/K+-ATPase also in cerebellar neurons?

7) Lines 81-84 discuss alterations in glycolytic metabolism; however, previously, emphasis was given to oxidative phosphorylation and the Krebs cycle as ATP production pathways. It is suggested to give a brief description of glycolysis before explaining, for example, that it decreases the activity of pyruvate dehydrogenase and hexokinase. 

8) Line 93, what would be the protective mechanism of coenzyme Q10 during the toxicity of monocytes with lead?

9) Please include the abbreviations of all figures, such as IAP, AIF, OMMP, etc.

10) DAMP was defined after its use, please correct. The following abbreviations were defined after their use:

PUMA

AKT

IAP

APAF

kDa

NLRP

Bcl

KFERQ

DRP

MFF

PGC

LC

ATM

COX

HSP60

TRIF

MAPK

IRF

NF-κB

ASC

AGER

TXB

THP

HR

10) The authors should discuss with more published works (in diseases) the role of lead in the mechanisms of mitochondrial dynamics and mitophagy in lead toxicity in the section called Lead-Induced Disruptions in Mitochondrial Dynamics and Mitophagy.

Author Response

Reviewer 2

The present review provides a detailed description of the molecular mechanisms that affect the dynamics, cell death processes, and alterations that lead to oxidative stress during lead exposure. It also mentions the processes involved and altered by lead in developing diseases, especially in tissues that strictly depend on mitochondrial function. The review is considered adequate for publication in this journal; however, some minor revisions could improve the quality and relevance of this work.

  • On line 42, you could add a paragraph describing the main and most common causes of lead poisoning.

We added relevant examples

  • In line 51 of point 2, please modify the sentence about metabolism efficiency since, at the end, it talks about glycolysis and oxidative phosphorylation, which are also metabolisms.

We modified the sentence accordingly

  • On lines 71-73, quotes need to support the information mentioned.

We added relevant quotes

  • Along these same lines, it is necessary to justify why emphasis is placed on the impact of lead on the nervous system and its components.

We reformulated the sentence accordingly

  • On line 78, what is the significance of AEC’s reduction during cerebellar neurons exposed to lead?

The adenylate energy charge (AEC) is an index used to measure the energy status of biological cells. According to Baranowska-Bosiacka et al. [12] high AEC ratios are typical of physiologically ‘healthy’ cells. In their study the mitochondria of the cerebellar granule cells isolated from Pb-treated rats demonstrated a significantly lowered concentration of ATP and AEC values. This suggests that under Pb toxicity conditions applied in their study, in neurons severe metabolic imbalance occurs.

  • On line 79, is the inhibitory effect of lead on Na+/K+-ATPase also in cerebellar neurons?

Yes, exactly. It has been shown in Baranowska-Bosiacka et al. study [12]

  • Lines 81-84 discuss alterations in glycolytic metabolism; however, previously, emphasis was given to oxidative phosphorylation and the Krebs cycle as ATP production pathways. It is suggested to give a brief description of glycolysis before explaining, for example, that it decreases the activity of pyruvate dehydrogenase and hexokinase. 

We are very sorry, but we do not understand what is the point of this remark. Therefore we are not able to give sufficient explanation.

  • Line 93, what would be the protective mechanism of coenzyme Q10 during the toxicity of monocytes with lead?

It has been proved that increased mitochondrial ubiquinone content results in a general improvement of bioenergetic parameters, like oxygen consumption, ATP content, mitochondrial potential and protein synthesis [Bergamini et al. PloS one. 2012 Mar 14;7(3):e33712 ]. We did not find any papers decribing the exact mechanism of its activity on monocytes under lead exposure. We can only speculate that this mechanism could be connected with a general improvement of monocyte energy metabolism, related to the action of Q10, what makes them less sensitive to toxic metals

  • Please include the abbreviations of all figures, such as IAP, AIF, OMMP, etc.

We included definitions of the abbreviations in the figures

10) DAMP was defined after its use, please correct. The following abbreviations were defined after their use:

PUMA

AKT

IAP

APAF

kDa

NLRP

Bcl

KFERQ

DRP

MFF

PGC

LC

ATM

COX

HSP60

TRIF

MAPK

IRF

NF-κB

ASC

AGER

TXB

THP

HR

We included definitions of the abbreviations in the text

  • The authors should discuss with more published works (in diseases) the role of lead in the mechanisms of mitochondrial dynamics and mitophagy in lead toxicity in the section called Lead-Induced Disruptions in Mitochondrial Dynamics and Mitophagy.

This topic has not been explored sufficiently enough. Available articles have been included in the present review